# Analysis of Small Area Environmental, Socioeconomic and Health Data in Collaboration with Local Communities to Target and Evaluate ‘Triple Win’ Interventions in a Deprived Community in Birmingham UK

**DOI:** 10.3390/ijerph16224331

**Published:** 2019-11-06

**Authors:** Patrick Saunders, Paul Campbell, Mark Webster, Michael Thawe

**Affiliations:** 1Faculty of Health Sciences, University of Staffordshire, Stafford ST18 0AD, UK; 2Birmingham City Council, 10 Woodcock St, Birmingham B7 4BL, UK; Paul.Campbell@birmingham.gov.uk; 3Welsh House Farm Big Local, 54 Rilstone Rd, Birmingham B32 2NR, UK; m.t.webster@icloud.com (M.W.); michael.thawe@gmail.com (M.T.)

**Keywords:** inequalities, environmental health, small area analysis

## Abstract

The contemporary environment is a complex of interactions between physical, biological and socioeconomic systems with major impacts on public health. It is well understood that deprived communities are more exposed to negative environmental and social factors, more susceptible to the effects of those exposures, more excluded from access to positive factors, less able to change their circumstances and consequently experience worse health, economic and social outcomes compared to the more affluent. Welsh House Farm estate in Birmingham is one of the most deprived areas in Europe. An alliance between a local charity, City Council Public Health and a University in collaboration with the local community has accessed, analysed and mapped a range of health, social and economic factors at small area level, identifying areas where the community experience is unacceptably worse than other parts of Birmingham and therefore requiring targeted interventions. We make specific recommendations for coordinated action addressing the living, moving and consuming domains of residents’ lives and have also identified positive aspects of life on the estate to celebrate. This pilot demonstrates the utility and cost-effectiveness of local collaboration to identify and target health, environmental and social inequalities informed by local concerns.

## 1. Introduction

Most of the great public health achievements in Europe have been due to improving physical and social environments rather than improved medical care [1], and the City of Birmingham, UK, is no exception. While our physical environment has improved dramatically over the last 100 years or so, it is clear that there is still much that can and should be done, and abundant evidence that this will deliver further major health dividends and reduce inequalities [2,3,4]. The nature and distribution of environmental stresses and their effects on communities and individuals have changed with new challenges emerging and old ones affecting us in unexpected ways; the recent re-emergence of air pollution as a significant public health issue is a case in point, highlighted by the 2016 Royal College of Physicians review [3]. There are other examples of course, and all are complicated by the interactions between environmental, biological and social systems. This ‘environmental health gap’, a lack of basic information needed to document links between environmental hazards and chronic disease, is one of the uncertainties that lead to widely differing estimates of the impacts in the research literature [2,5,6], but what is certain is that the wider and local environments have major negative and positive consequences for health and well-being. It is also evident that stressors do not operate in isolation [1] and that access to health-improving environments such as good quality green space is inequitably distributed [7]. Deprived communities are almost invariably not only more exposed to environmental, social and public health pressures [4,5], they experience multiple exposures and inequalities at the same time, and are more susceptible to the effects of those exposures [3]. This reflects the triple jeopardy recently referred to by the Chief Medical Officer for England [8]. Effectively addressing one stressor can also address others e.g., developing programmes that encourage local sourcing and community production of healthy foods can improve dietary health, environmental sustainability through reducing ‘food miles’ and reduce inequalities; the ‘triple win’ that underpins the EU INHERIT programme [9]. Identifying which stressors are the most important and describing their distribution is critical for effective action to protect people and communities. Here, the concerns of the professionals and the public appear to diverge. Environmental Health research has understandably focused on large-scale issues such as climate change rather than more immediate local impacts. However, when asked which environmental health issues are most important to them, local people invariably raise matters of basic environmental amenity, such as litter, fly-tipping, noise, bonfires, housing disrepair, street lighting and derelict land [10,11,12]. These are not, of course, mutually exclusive domains and actions to address one will often contribute to addressing the others. These public health nuisances are often considered as simply irritations or ‘quality of life’ issues but in reality, have a direct and significant impact on health and well-being [13] and reflect the daily lived experiences of local people and communities. The Victorians recognised this as an indefensible injustice as well as a health issue and established a great body of public health and social welfare law to tackle it and local authorities to enforce those laws. Enforcement of this legislation has had a huge impact on improving people’s health and quality of life and retains the potential to be even more influential. However, to date, enforcement has been almost entirely reactive and there has been little research into the distribution of public health nuisances or the potential for tailoring interventions to reduce inequalities in exposure and effects or to act to prevent nuisances in the first place. Community input to decisions about priorities is important in terms of maximising public benefit and strengthening the relationship between communities, public health professionals and politicians. This collaboration between Birmingham City Council, a National Lottery funded community group Welsh House Farm (WHF) Big Local Project [14], and a local Public Health academic was established in 2017 to develop a pilot project in the WHF housing estate to explore the potential of small area data mapping and analysis together with community consultation. The estate, completed in 1965, is about four miles from Birmingham City Centre with a population of approximately 3500 residents, almost half of whom live in social housing including tower blocks. The estate is described in the Birmingham Unitary Development Plan as ‘an isolated area of poor quality housing in need of improvement’ [15]. It is an intensely deprived area. The Index of Multiple Deprivation 2015 is the official measure of relative deprivation for small areas (or neighbourhoods) in England [16]. Of the two Lower Super Output Areas (LSOA), base UK census geographies that cover the estate (see Figure 1) indicates one is in the most deprived 2% and the other in the most deprived 5% of the 32,000 LSOAs in England. This paper describes the collaboration’s work to establish a system for the routine monitoring, integration and analysis of data to identify key areas and issues for interventions to protect health and reduce inequalities taking the local community’s as well as the professionals’ concerns into account. This is especially important in an era of public sector austerity where local authorities must focus their resources in the most cost-effective way. This pilot has demonstrated the project’s utility and it is currently being considered for application to the entire City and will be extended to include other key issues in collaboration with local communities.

## 2. Methods

The data and other intelligence on local environmental and social health, the statistical expertise to analyse these data, and the power to intervene to tackle problems do not lie with a single agency. This programme requires a real alliance of community workers, the community itself, academics, and the statutory bodies. The first step was to establish this network. The WHF Big Local has deep roots in the local community, an understanding of its needs and a mandate to improve health and social well-being and cohesion. This is reflected in its 2017 action plan, in particular, the establishment of an Environmental and Gardening group, litter picking, targeting areas for improvement, supporting families in need, community listening, health and well-being initiatives and engaging with professionals [17]. As a result, the Big Local has invested time and energy in building a sound relationship with local authority colleagues and has access to academic skills through it’s volunteering network. This came together in 2017 at a very successful and well-supported Big Local Health Day attended by Birmingham City Council’s Director of Public Health and the subsequent establishment of a small collaborative working group.

For the pilot to be practical and deliverable within a reasonable timescale, it was necessary to agree to a small set of relevant issues for which data were available, readily accessible, of adequate quality to enable meaningful analysis and not requiring a lengthy ethical approval process for release. The working group identified the following:Nuisance complaints made to the local authority (air pollution, noise, animals, rubbish dumping) 2014–2017;Housing-related complaints and issues (forced entry and/or evictions, filthy and verminous premises, infestations) 2014–2017 and reactive council housing repairs 2016–2018;Health (mental health outpatient attendances, drug and alcohol-related outcomes including hospital admissions,) varying time periods;Social (anti-social behaviour including crime and alcohol-related incidents, EU and non-EU migration, car availability, benefit claimants, free school meals).

These criteria reflect five of the ten Building Research Establishment Healthy Cities categories [18]. Sources of data are given in Table 1:

The smallest area for which Office for National Statistics (ONS) population data are available is the Lower Super Output Areas (LSOA) and data for these were extracted for analysis including the two LSOAs (combined population 3416) closely matching the Welsh House Farm (WHF) estate (see Figure 1) which we assessed in relation to all LSOAs within Birmingham.

Complaint rates per 10,000 population were calculated for nuisance and housing complaints and forced entry/evictions and compared with the rates for Birmingham (population = 1.1 million). Rates per 1000 population were calculated for anti-social behaviour, total crime, Job Seekers Allowance (JSA) claimants and migration, and per 100 population for car or van availability. Alcohol-related outcomes were calculated as crude rates except alcohol-specific deaths and liver disease preventable deaths (directly age-standardised rate to account for the distorting effect of a disproportionately young or old population). The percentage of children receiving free school meals was calculated.

A key element of this project is to identify issues in the WHF area that are so much more challenging than the rest of Birmingham as to require investigation and potential intervention to address. Statistical Control Charts (SCC) are widely used in industrial quality control and in health research to highlight the key issues from the ‘background noise’ and were used to interrogate nuisance (noise, pollution, rubbish and dog complaints), housing, anti-social behaviour, car availability, JSA claimants, children receiving free school meals, migration, infestations and crime. An initial assessment of local inpatient and outpatient data showed such unpredictability they could not be included. 

Controls limits were set at two standard deviations (SD) and three SDs, points outside the latter being highly significantly different than expected.

## 3. Results

There were 10,061 events in the LSOAs combined and 715,717 for the City were included in the analysis.

### 3.1. Social

The SCC analyses show a significantly high rate of anti-social behaviour (2016–2017) at two SDs in one LSOA and a rate within the expected range for the other, although levels of total recorded crime are either at or below the average for the City (see Figure 2 and Figure 3; the WHF LSOAs are shown throughout in charts in red). Access to a car or van is significantly below the City average in one LSOA and highly significantly below in the other (Figure 4). Levels of inward migration from abroad are at the City average or highly significantly below the average (Figure 5). The rate of JSA claimants in both LSOAs is highly significantly above the City average (see Figure 6) as is the number of children receiving free school meals at the local school (see Figure 7), reflecting the levels of deprivation in WHF.

### 3.2. Health

The rates of outpatient attendances under mental health specialty doctors are highly significantly elevated in both LSOAs (Figure 8). The rates of alcohol-related admissions, alcohol-related safeguarding issues and numbers in the City’s commissioned services for drug and alcohol misuse are elevated in both LSOAs, and the rate of non-domestic violent incidents involving the police was elevated in one of the LSOAs. The remaining alcohol outcomes were similar to or below the levels experienced on average in the City.

### 3.3. Nuisance and Housing Related Complaints 

The monthly rate of complaints (April 2014–March 2017) relating to dogs in WHF was highly variable, which is unsurprising given the randomness inherent in the relatively small numbers involved. The SCC analysis showed the overall complaint rate in one of the WHF LSOAs to be within the expected range but highly significantly elevated in the other (Figure 9). There is a similar pattern with infestation complaints, with considerable variability across the three years and some very periodic high rates, although the SCC shows there are no exceptional levels of complaints (Figure 10). The air pollution complaint data show a significantly elevated overall level in one LSOA at two SDs and a much lower rate within the expected range for the other using SCC analysis (Figure 11). It is important to note that these data relate to *complaints* rather than measured levels of air pollution, which would be expected to increase with increasing population density.

Levels of noise complaints were highly significantly elevated in both WHF LSOAs (Figure 12) as were rates of filthy and verminous premises complaints (Figure 13). Forced entry/evictions showed a sharp difference between the two LSOAs with a highly significant level in one LSOA and a significantly lower level in the other (Figure 14). Reactive council housing repairs in both LSOAs were well within the expected range and the higher level in one LSOA (Figure 15) would be expected given the higher number of local authority properties.

Rubbish related complaint rates generally follow the trend for Birmingham and are within the expected range in one LSOA and significantly lower than expected in the other (Figure 16).

Overall, results differed between the two LSOAs for seven indicators, including cases where one LSOA was highly significantly elevated and the other highly significantly lower than expected (forced entry evictions), within the expected range for one LSOA and highly significantly worse than expected in the other (dog-related complaints), and for rubbish related complaints which were in the expected range in one and highly significantly lower in the other LSOA (see Table 2).

## 4. Discussion

This pilot project has revealed some concerning, some positive, and some intriguing environmental and social aspects of life in WHF. Some results are so in excess of the City average to warrant further investigation to identify their underlying causes and to target interventions to reduce their levels and/or prevent their reoccurrence. The levels of dog-related complaints, for example, show both an extended period of unacceptably high public health nuisance and periodic exceptional peaks. The definition of such complaints is wide and could include noise, fouling, stray animals and/or aggressive dogs. Interestingly, and perhaps related, the levels of noise complaints are similarly of concern. Whatever the specific causes, all of these events adversely affect individual and community safety, well-being, cohesion, access to community resources and, in extreme cases, physical health. Local intelligence, albeit anecdotal, suggests that inadequate and inappropriate siting and use of dog fouling bins for hygienic disposal could plausibly be one cause. 

Levels of anti-social activity are significantly higher in WHF than other parts of the City. Such activity is hugely damaging to communities reflected in the personal experiences and stories of residents and a burden on local services including the Police, local authority, primary care, and social services. There are also adverse effects on many of the perpetrators in terms of their future lifestyles and life chances. The levels of mental health outpatients’ attendances are worryingly and significantly higher than the City average. However, this finding needs to be interpreted with some caution as the analysis has not been controlled for age or distinguished between minor and severe conditions, and the data could be skewed by a small number of people attending multiple times. Nonetheless, this is a concerning signal and requires further exploration including working with local clinicians. Following consultation with local residents, a community-led mental health support group has been established with professional support from the City Council which meets monthly enabling people to talk freely about their issues in an informal environment.

While the number of forced entry evictions is relatively small, these actions are associated with potentially devastating consequences for the victims, both those evicted and those neighbours who, on occasions, will be the subject of the behaviours leading to the evictions. Prevention and multi-agency cooperation are absolutely critical. Of similar concern, are the elevated levels of alcohol-related hospital admissions, safeguarding issues and violence in WHF.

Although the overall trend of filthy and verminous premises complaints closely follows that of the City average, there are concerning periodic peaks. This type of public health nuisance is often associated with other social stressors and vulnerable persons and families and so is particularly important for the local authority and the community to address and prevent. The latter potentially plays a vital role in alerting and supporting both the former and the affected people. 

The local authority needs to examine the data for all the above in more detail to distill out the key potential causes and work with the local community to develop appropriate remedial preventative actions. These include more targeted Environmental Health Practitioner and Police responses and inspections. The latter needs to be considered in light of the closure of the local Police Station due to austerity cuts in funding.

There are sharp differences in some measures between the two WHF LSOAs in terms of forced entry evictions, anti-social behaviour, and complaints about rubbish, air pollution and dogs. It is likely that are some specific local environmental or behavioural factors contributing to these differences. It is important to identify these differences and ensure that people living in the poorly performing LSOA enjoy a similar level of environmental and social quality as their neighbours. 

The topography of the WHF estate which includes narrow roads, sharp turns and dips means that polluting buses and other vehicles are routinely idling, reversing and queuing particularly at certain times of the day. The close proximity of dwellings and the school to the road creates a major exposure risk to a community which, because of its overall low levels of car ownership, actually contributes little to creating that risk. This is a gross inequity. Car ownership, a proxy measure of deprivation, is significantly below the City average and in one LSOA, highly significantly so. Increasing car ownership is not compatible with improving health or the environment. In addition, there is good evidence that poorer people are more vulnerable to adverse effects of air pollutants [8] and this is a community that is severely deprived as shown by the level of JSA claimants and children receiving free school meals. The community deserves reliable data on local air quality to both enable residents to avoid high levels of pollution and also to empower them to demand change. A low-cost monitor for particulate matter, one of the most dangerous air pollutants, has been developed by scientists based at Birmingham Children’s Hospital for deployment at schools [19] and this opportunity should be seized in partnership with the local school. A marked increase in noise and pollution complaints in the summer coincided with community intelligence about large-scale noisy barbecues held on communal green areas. This presents the opportunity for the statutory agencies to intervene early to prevent such nuisances and, perhaps even more importantly, work with residents to ensure that their use of the available green resources is encouraged in a way that benefits all users and residents. The current work of the Big Local Project in terms of increasing opportunities for physical activity and urban gardening is laudable and should be expanded, but communities cannot be excluded from accessing essential public and commercial services and facilities simply because they are poor. In this light, improvements to local bus services and their costs are essential. In addition, there is very limited opportunity for residents to source fresh fruit and vegetables on the estate, and the collaboration is working with an allotment in an adjacent ward to provide such produce to the local food bank, and also on a programme to encourage WHF residents to take on allotment pitches to grow their own.

Local politicians report that many residents are concerned about high levels of migrants being housed in WHF. The data clearly show that is not the case, the opposite in fact, and while there can be no pandering to racist, sectarian or xenophobic myths, local concerns about the scale of migration can be readily addressed. Indeed, given that these people will be amongst our most vulnerable, the agencies must work together to ensure that they are not disproportionately exposed to the stressors described above.

It is also especially important to recognise that for some metrics, WHF is actually ‘better’ than average and this opportunity to tell a ‘good story’ should not be missed. Establishing the reasons why WHF is performing well in these areas will provide useful evidence for other parts of the City.

It is important to recognise that this analysis covers a relatively short time period and involves relatively small sample sizes for some metrics which introduces the risk of statistical uncertainty. However, this preliminary analysis of a sample of routinely available data has demonstrated the potential of a partnership between the voluntary sector, the City Council, academics and the community to share resources, skills and knowledge to assess the scale, distribution and impacts of the physical and social environment on the health and quality of life of a local community. The project partners will develop an action plan to address the issues and opportunities identified and the City Council is currently considering extending the system beyond WHF to include a broader range of environmental and social issues and related health outcomes.

## 5. Conclusions

This pilot has used, for the first time at this level to the authors’ knowledge, SCC to target routine nuisance inspections and to routinely monitor the relationship between hazards and disease. This has delivered both concerning and reassuring outputs, the former providing intelligence for intervening, the latter providing positive messages for residents and for the image of the community as a whole. This pilot has provided the basis for the development of a wider Environmental Public Health Tracking programme at a marginal cost [20]. It is increasingly evident that we are simply not able to deliver improved and equitable standards of health, wellbeing and health care in the medium to longer-term without, as a society, paying much more attention to the physical and social environments. Tracking has the utility to both address local environmental, social and health issues and to contribute to action across the City required for long-term sustainable public health improvements. 

## Figures and Tables

**Figure 1 ijerph-16-04331-f001:**
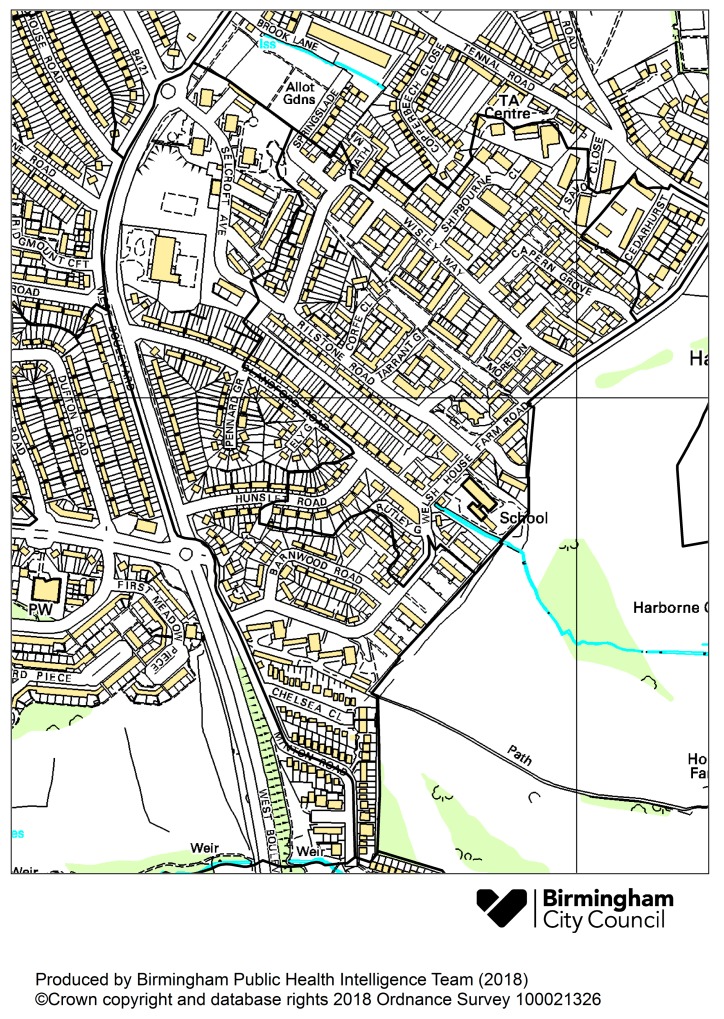
Lower Super Output Areas approximating to WHF Estate.

**Figure 2 ijerph-16-04331-f002:**
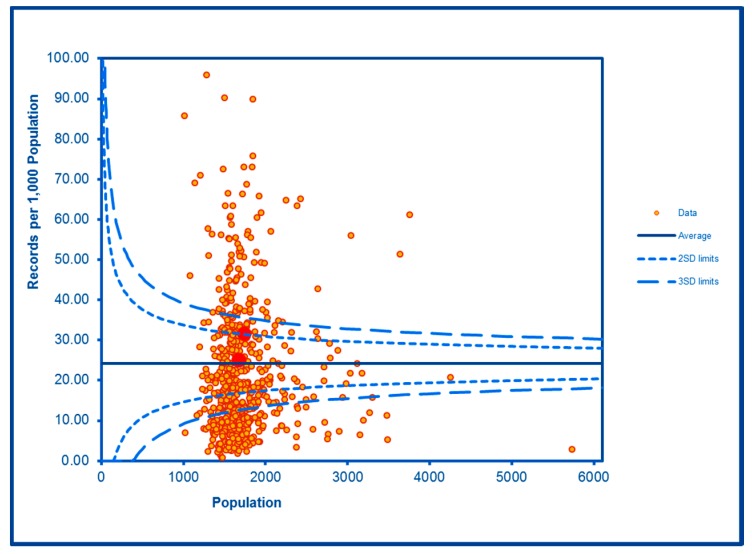
Control Chart for Anti-Social Behaviour Incidents.

**Figure 3 ijerph-16-04331-f003:**
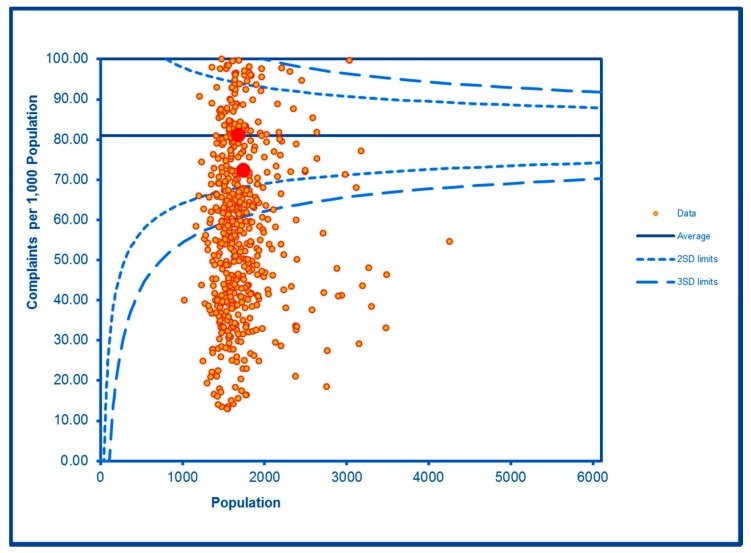
Control Chart for All Crimes.

**Figure 4 ijerph-16-04331-f004:**
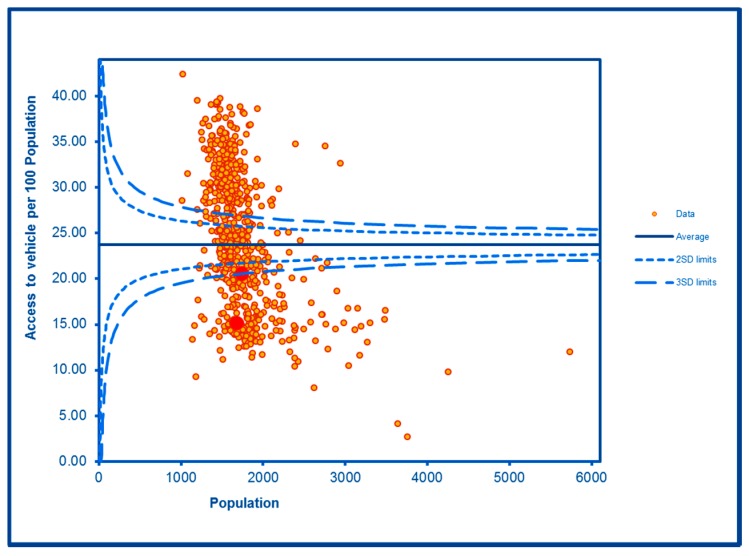
Control Chart for Household Car or Van Availability.

**Figure 5 ijerph-16-04331-f005:**
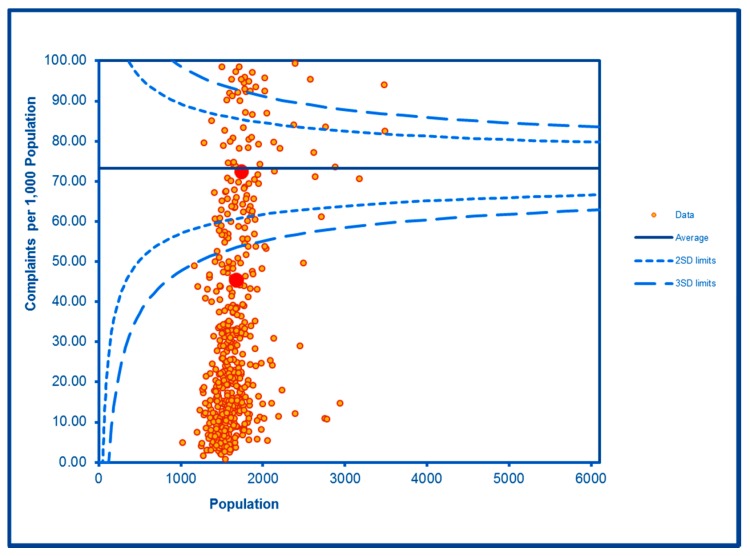
Control Chart for Migrants Housed.

**Figure 6 ijerph-16-04331-f006:**
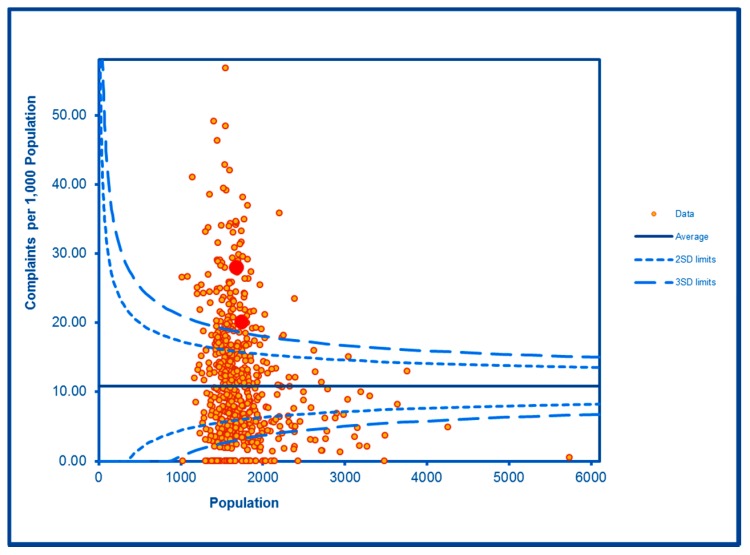
Control Chart for JSA Claimants.

**Figure 7 ijerph-16-04331-f007:**
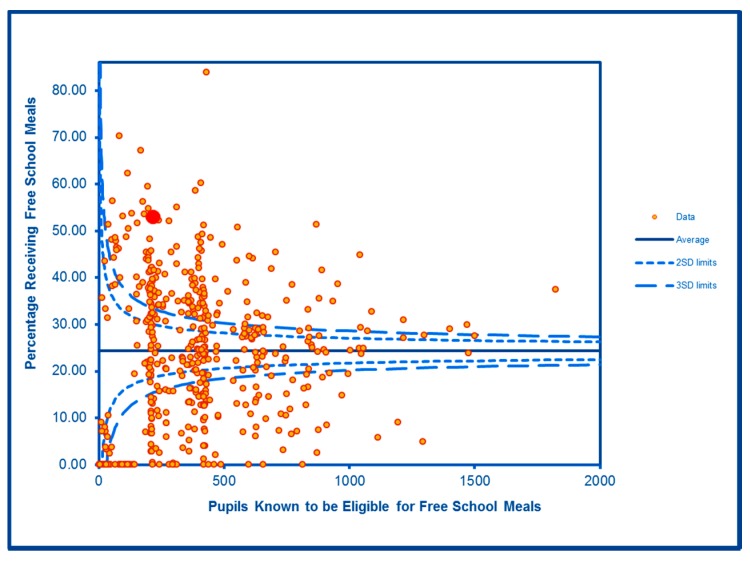
Control Chart for Proportion of Children Receiving Free School Meals.

**Figure 8 ijerph-16-04331-f008:**
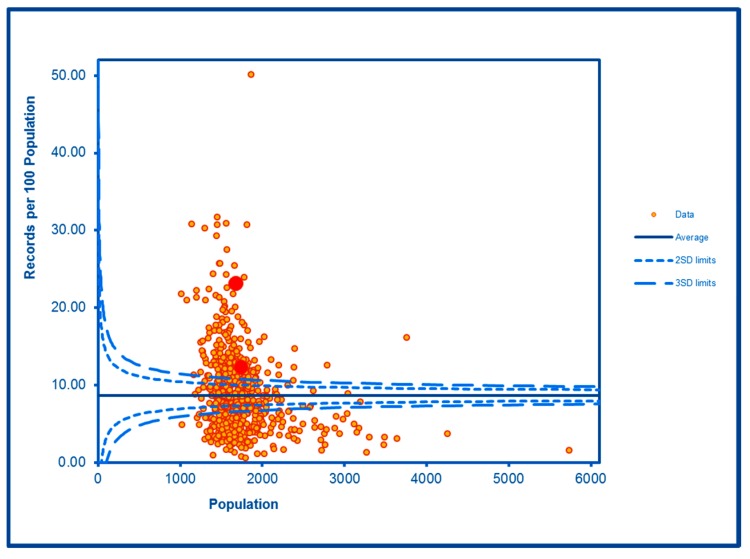
Control Chart for Mental Health Outpatient Attendances.

**Figure 9 ijerph-16-04331-f009:**
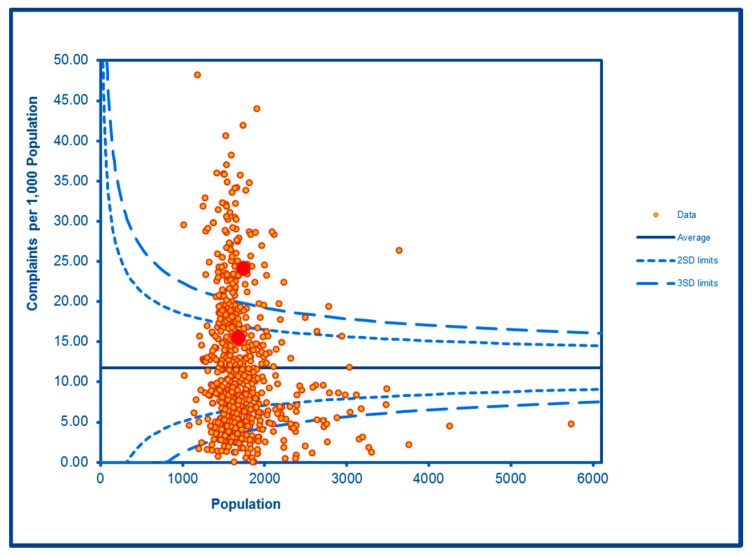
Figure 8 Control Chart for Dog-Related Complaints.

**Figure 10 ijerph-16-04331-f010:**
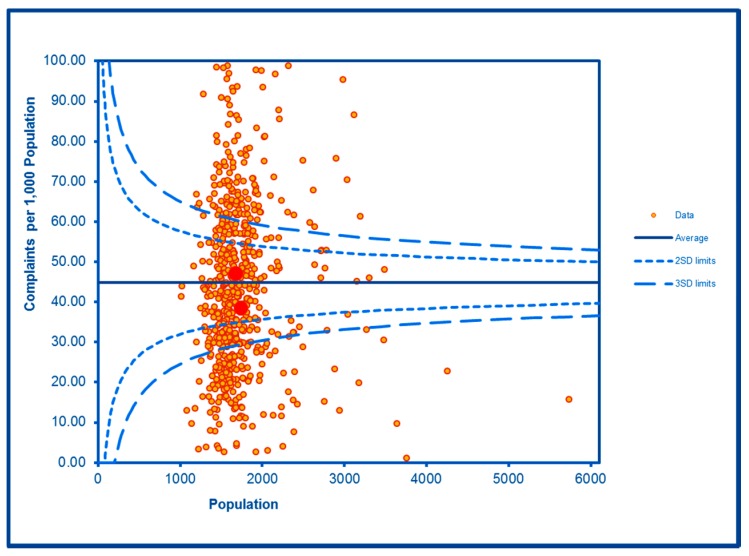
Control Chart for Pest (vermin etc) Complaints.

**Figure 11 ijerph-16-04331-f011:**
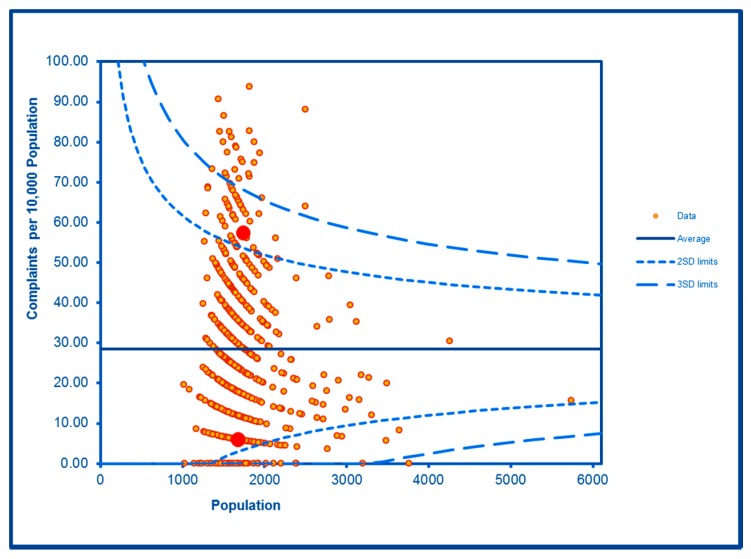
Control Chart for Air Pollution Complaints.

**Figure 12 ijerph-16-04331-f012:**
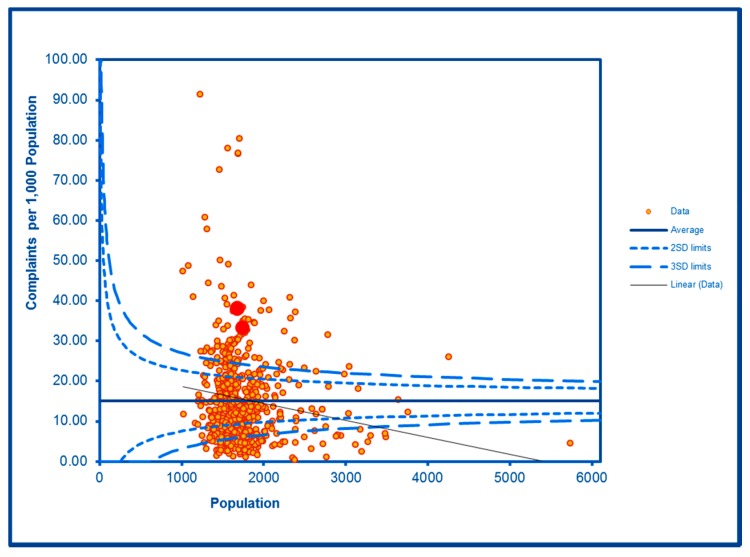
Control Chart for Noise Complaints.

**Figure 13 ijerph-16-04331-f013:**
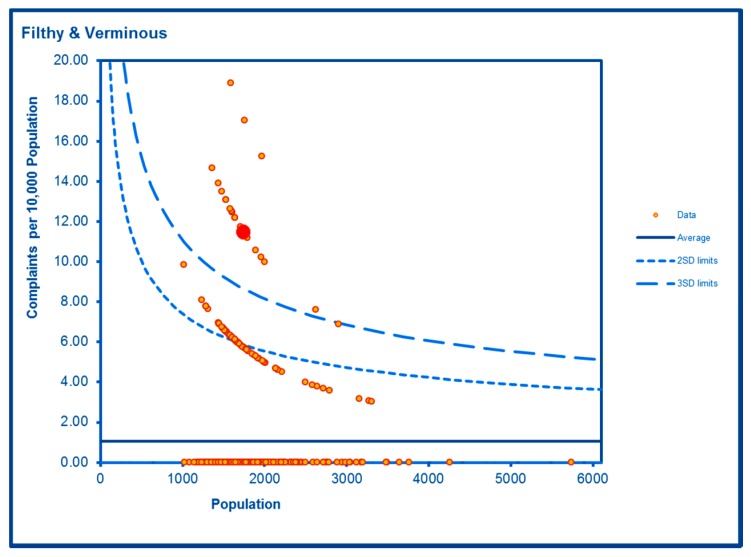
Control Chart for Filthy and Verminous Premises Incidents.

**Figure 14 ijerph-16-04331-f014:**
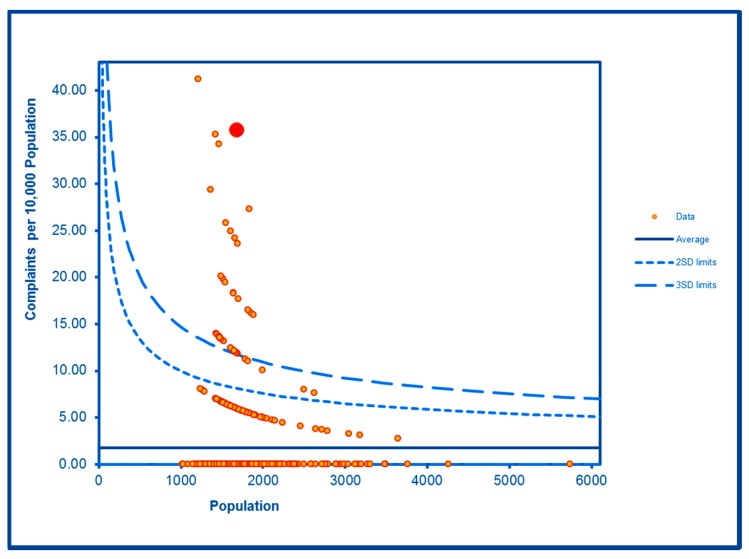
Control Chart for Forced Entry Evictions.

**Figure 15 ijerph-16-04331-f015:**
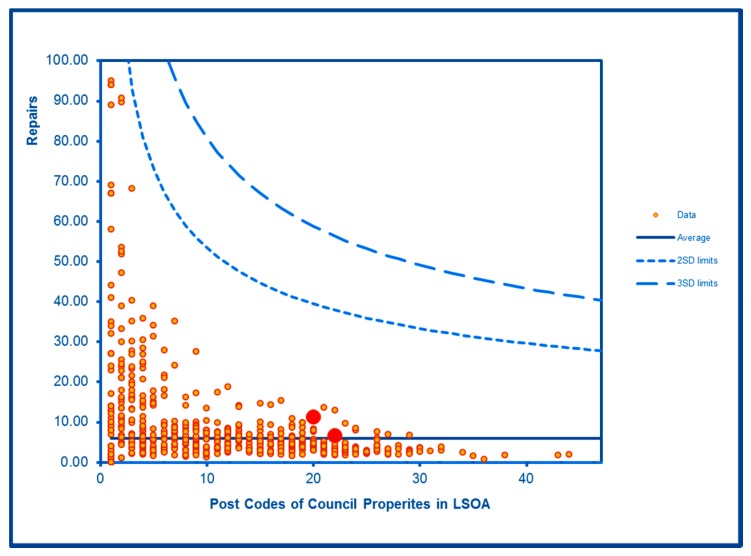
Control Chart for Council House Repairs.

**Figure 16 ijerph-16-04331-f016:**
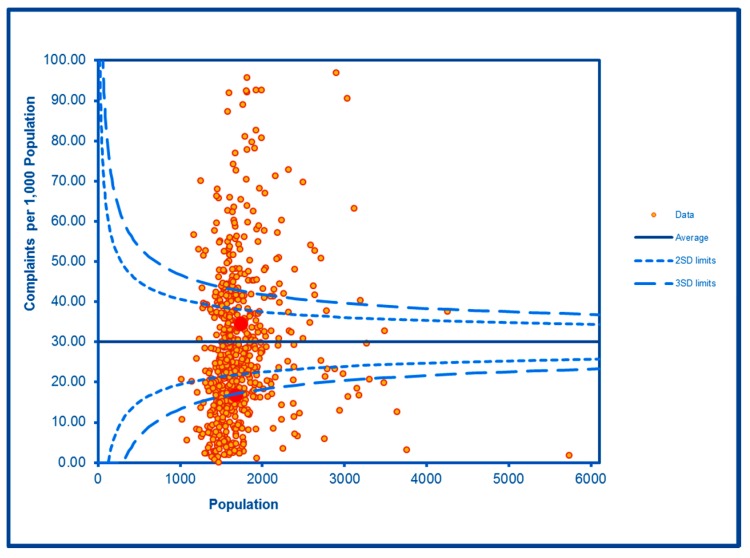
Figure 12 Control Chart for Rubbish Related Complaints.

**Table 1 ijerph-16-04331-t001:** Sources of data.

Indicator	Data Source
Nuisance complaints	Birmingham City Council
Housing-related complaints and reactive council housing repairs	Birmingham City Council
Drug and Alcohol-related outcomes	Birmingham City Council, West Midlands Police, West Midlands Fire Service, the Office for National Statistics (ONS), Public Health England, Commissioned Support Services
Health	Hospital Episode Statistics, NHS Digital
Social	Migration Observatory, West Midlands Police, ONS, Official Labour Market Statistics (NOMIS), Department for Education

**Table 2 ijerph-16-04331-t002:** SCC results by LSOA (bold indicates significance at above or below three SDs).

Description	West LSOA E01009073	East LSOA E01009074
Anti-Social Behaviour	Higher than expected	Expected
Car Availability	**Lower than expected**	Lower than expected
Migration	**Lower than expected**	Expected
Council Housing Repairs	Expected	Expected
Total Recorded Crime	Expected	Expected
Air Pollution	Expected	Higher than expected
Forced Entry Eviction	**Higher than expected**	**Lower than expected**
Dog-related	Expected	**Higher than expected**
Filthy and Vermin	**Higher than expected**	**Higher than expected**
Noise related	**Higher than expected**	**Higher than expected**
Pests	Expected	Expected
Rubbish Related	**Lower than expected**	Expected
MH OP Attendances	**Higher than expected**	**Higher than expected**
JSA Claimants	**Higher than expected**	**Higher than expected**
Free School Meals	**Higher than expected (WHF school)**	

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
