# Peer review of "Analysis of Small Area Environmental, Socioeconomic and Health Data in Collaboration with Local Communities to Target and Evaluate ‘Triple Win’ Interventions in a Deprived Community in Birmingham UK"

_ijerph, 2019, doi:10.3390/ijerph16224331_

Round 1
Reviewer 1 Report
This study presents the results of a community survey conducted in Birmingham, UK, in regard to the living environment in a neighborhood. The survey collected a variety of neighborhood characteristics ranged from crimes to air pollution complains. This study can provide a reference for urban planning in the neighborhood. The manuscript is well-written. However, several key steps are missing, that makes this manuscript difficult to follow. Detailed comments are here:
What are the characteristics of the “Lower Super Output Area”? Is it a low-income neighborhood? How to define the “expected” value of survey results? What would be the reference group? What are the big red dots in most figures? In figure 11, it seems the complaints of air pollution decreased as the increase of population. But generally, more populous areas have worse air pollution problems. So I’m wondering if the author can show the air pollution complains by population density (people/km2)? Or make a discussion about this inverse association between air pollution complains and population? In the Discussion, the author mentioned “periodic peaks” several times. But there are no periodic results reported in the Result section. I suggest focusing on the results presented in the manuscript.
Author Response
Thank you for your comments. I have amended the introduction to include a more detailed description of LSOAs and the levels of deprivation in the estate and have added two references. I have added a statement in the methods to clarify the reference group. The 'red dots' relate to the results for the two LSOAs of interest and I have amended the text to clarify. In regards to air pollution I have amended the text to clarify that these data relate to complaints about air pollution rather than measured ambient levels and have added a statement about the increases in the latter normally being associated with increased population density. Finally I have deleted the discussion about 'periodic peaks'.
Reviewer 2 Report
The paper is a pilot study on understanding a real relationship between exposition to negative environmental and social factors and effects on population that live in a deprived area. The research takes into consideration different variables (at large scale) and methodology proposed is suitable for handling the data a for giving robust findings.
The study is suitable for publication even if a more detailed description of the Welsh House Farm estate (that is the object of the study). The authors underlined that it is one of the most deprived area in Europe and some information is reported in the Introduction but it is not sufficient to understand the characteristics of the area, especially in according to the research aims!
I suggest to introduce a brief paragraph where describing the area and the community with more details.
Author Response
Thank you for your comments. I have added a more detailed description of the Welsh House Farm Estate in the introduction
Reviewer 3 Report
Dear Authors
I congratulate you on your clear, objective and relevant article. It was very enriching to read it although the theme is not new.
But its never to much to sensitize and investigate on this subject. In addition to being a crosscutting theme to society you have done it with rigor and detailed methodology and didn't forgot all the actors including recommendations to the the local government.
It emphasizes the importance of cumulative and simultaneous development of cognitive and social capacities in establishing a new relationship with the environment.
I hope you continue to explore the topic.
Best regards
Author Response
thank you for your generous comments
Round 2
Reviewer 1 Report
All of my comments and questions have been addressed in the revision. Thanks.